# Reinforcement Learning for Tool-Calling Agents in Fast Healthcare Interoperability Resources (FHIR)

**Marius S Knorr** [* 1]  **Robert Müller** [* 2]  **Jan P Bremer** [1]  **Nils Schweingruber** [1]

## Abstract

Fast Healthcare Interoperability Resources (FHIR) is the dominant standard for interoperable exchange of healthcare data. In FHIR, electronic health records form a directed graph of resources. Answering clinically meaningful questions over FHIR requires agents to perform multi-step reasoning, filtering, and aggregation across multiple resource types. Prior work shows that even tool-augmented LLM agents (retrieval, code execution, multi-turn planning) often select the wrong resources or violate traversal constraints. We study this problem in the context of FHIR-AgentBench, a benchmark for realistic question answering over real-world hospital data, and frame reasoning on FHIR as a sequential decision-making problem over a queryable structured graph. We implement a multi-turn CodeAct agent and post-train it with reinforcement learning using a custom harness and tools. A LLM Judge provides execution-grounded rewards. Compared to prompt-based, closed-model baselines, RL post-training improves performance while enforcing data-integrity constraints. Empirically, our approach improves answer correctness from 50% (o4-mini) to 77% on FHIR-AgentBench using a smaller and cheaper Qwen3-8B model. We present an end-to-end post-training pipeline (environment building, harness construction, model training and custom evaluation) that reliably improves multi-turn reasoning over structured clinical graphs.

## 1. Introduction

The broad availability of electronic health data (Adler-Milstein et al., 2017; Kim et al., 2024) holds potential to enable a wide range of data-driven applications by integrating signals across different data modalities. A primary barrier to realizing this potential is not only the high dimensionality and incompleteness of EHR (electronic health records) data, but its lack of standardization across institutions, vendors, and workflows (Reisman, 2017). As a result, leveraging clinical data at scale frequently requires site-specific schema alignment through one-off ETL pipelines, limiting reproducibility and portability.

To address interoperability, the health IT ecosystem has increasingly converged on Fast Healthcare Interoperability Resources (HL7 FHIR) (Health Level Seven International, 2019), a data model for clinical and administrative data with a RESTful web API for exchange and access. In FHIR, entities are represented as resources (e.g., *Patient*, *Encounter*, *Observation*, *MedicationRequest*) that may reference one another. Consequently, a patient chart can be viewed as a directed graph with nodes and edges.

Clinically meaningful information is rarely captured in a single resource but instead distributed across several linked nodes. For example, the fact that a patient was prescribed a particular medication and later showed elevated creatinine spans separate *MedicationRequest* and *Observation* resources connected through shared references to an *Encounter* and *Patient*. Making sense of such data therefore requires multi-hop traversal across these resources, together with operations such as filtering or temporal sorting. This turns clinical question answering into a graph navigation problem.

Lee et al. (2025) target this challenge with FHIR-AGENTBENCH, a benchmark of 2,931 clinician-sourced questions grounded in real de-identified patient records from MIMIC-IV converted to FHIR format. The benchmark evaluates whether LLM agents can navigate the FHIR graph, retrieve the relevant data, and reason over it to produce correct answers. Their systematic evaluation compares retrieval strategies (direct FHIR API calls vs. specialized tools), interaction patterns (single-turn vs. multi-turn), and reason-

---

[*]Equal contribution  [1]IDM gGmbH, University Medical Center Hamburg-Eppendorf, Hamburg, Germany  [2]Aganthos. Correspondence to: Marius Knorr <knorr@idmedizin.de>.

*Proceedings of the 43rd International Conference on Machine Learning*, Seoul, South Korea. PMLR 306, 2026. Copyright 2026 by the author(s).

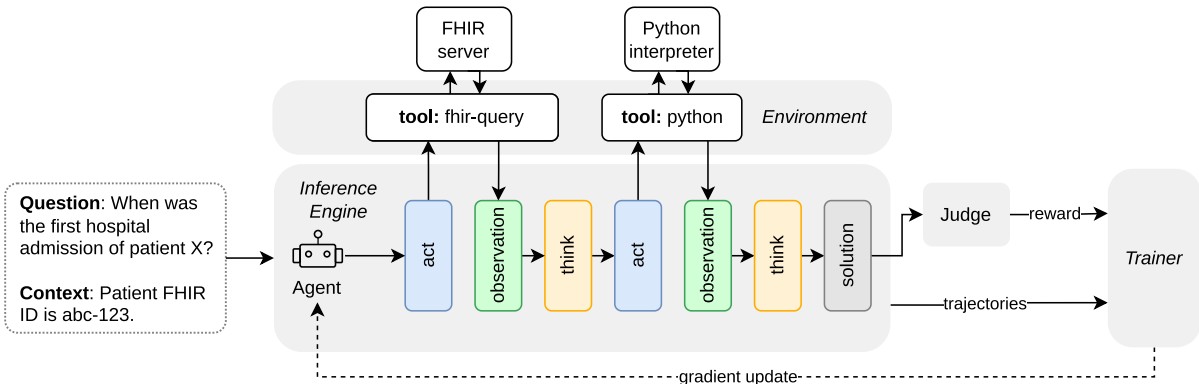

*Figure 1.* Reinforcement learning pipeline for training a clinical FHIR reasoning agent. The agent follows a structured act–observe–think loop, with access to two tools: a retrieval tool for loading clinical resources from a FHIR server, and a Python interpreter for code execution. After one or more reasoning cycles, the agent emits a final solution, which is evaluated by a LLM-judge against ground-truth answers from FHIR-AgentBench to produce a reward signal. The rewards and full interaction trajectories are then fed to the trainer, which updates the agent's policy via GRPO, closing the RL loop.

ing strategies (natural language vs. code generation). The results expose two core bottlenecks: on the retrieval side, agents select wrong resource types or miss required traversal steps; on the reasoning side, they mishandle temporal logic, case-sensitive terminologies, or multi-step aggregation over the retrieved data. Even in their strongest configuration (o4-mini with multi-turn planning and code execution), accuracy reaches only approximately 50%, showing the gap between current agent capabilities and the demands of structured clinical reasoning over FHIR.

We hypothesize that this brittleness stems from a gap between FHIR's standardized interfaces and real-world deployments. In practice, optional fields are populated inconsistently, and the same clinical concept can be encoded in different resources or profiles. Prior work has shown that the use of *Observation* is particularly inconsistent across implementation guides, though this heterogeneity extends beyond *Observation* (Kramer, 2023). Single-shot LLMs that assume an idealized schema therefore struggle when faced with real-world FHIR data. In contrast, a multi-turn interaction pattern can support schema discovery: the agent can inspect candidate resources, verify how resources are stored, and iteratively refine queries before committing to an answer (Lee et al., 2025).

To operationalize this idea, we adopt a CodeAct-style agent paradigm (Wang et al., 2024) in which actions are executable programs that query the FHIR API, inspect intermediate results, and self-correct across turns. We implement this multi-turn code agent within SkyRL, a post-training harness that supports both evaluation of open-weight models and specialization for FHIR QA through execution-based rewards (Figure 1).

Building on FHIR-AGENTBENCH (Lee et al., 2025), we

show that post-training with multi-turn tool use enables the agent to internalize the specific structure and conventions of the underlying FHIR server through repeated interaction and debugging.

Empirically, we improve accuracy from 50% (o4-mini) to 77% using a substantially smaller Qwen3-8B model. We further provide a practical post-training recipe and an analysis of failure modes, highlighting the remaining gaps toward reliable FHIR-based clinical QA.

## 2. Background

### 2.1. FHIR

Fast Healthcare Interoperability Resources (FHIR) is an HL7 standard for representing clinical and administrative data as a collection of typed resources (e.g., *Patient*, *Encounter*, *Observation*, *MedicationRequest*) exchanged via a RESTful API. Each resource is a self-contained JSON document identified by its type and a unique ID, and contains nested, domain-specific attributes. Resources reference one another through typed Reference fields. For instance, an *Observation* points to the *Patient* it describes, forming a typed, directed, heterogeneous graph (Figure 2) that can be traversed and queried incrementally.

From an ML viewpoint, a patient record is therefore not a single table but a graph of heterogeneous nodes with sparse, inconsistently populated attributes; answering a clinical question requires programmatic traversal and computation over this structure.

FHIR standardizes resource types and interaction patterns but intentionally leaves room for local specialization. Deployments routinely constrain base resources through *pro-*

*files* (*StructureDefinition*), introduce site-specific fields via *extensions*, and adopt different terminology systems for the same clinical concepts. This flexibility is essential for adoption, but it creates a core ML challenge: two sites can represent the same clinical fact using different resource types, fields, or coding conventions, while optional elements may be populated inconsistently.

## 2.2. Related Work

We find it useful to separate prior work into two pipelines: *(i) populating* a FHIR server (mapping raw clinical data into FHIR resources), and *(ii) reading* a FHIR server (retrieving/traversing resources to answer questions or complete tasks).

*(i)* A line of work uses LLMs to convert clinical narratives or structured datasets into FHIR-compliant resources. Li et al. (2024) propose FHIR-GPT, targeting medication-focused extraction by transforming free-text snippets into *MedicationStatement* resources and evaluating against an annotated dataset. More recently, Riquelme et al. (2025) study LLM-assisted transformation of structured clinical tables into FHIR on MIMIC-IV, using retrieval and schema-aware prompting to map tabular fields to resource attributes. Complementary to these model-centric efforts, Idrissi-Yaghir et al. (2025) introduce FHIR-Workbench, a suite of datasets that includes note-to-FHIR generation and resource recognition tasks, providing standardized evaluation for both write and read capabilities.

*(ii)* A different line of work treats a FHIR store as the backend for question answering or interactive clinical agents. Schmiedmayer et al. (2024) built an application that lets users query FHIR-formatted patient data with an LLM through a mobile application (synthetic data). Kothari & Gupta (2025) follow a retrieve-then-read pipeline: they first retrieve relevant FHIR resources for a query and then answer using a fine-tuned (private) LLM, focusing on patient-record QA with privacy constraints (using synthetic data via Synthea). Benchmarks have recently matured from static QA sets to interactive, tool-using evaluations. Jiang et al. (2025) introduce *MedAgentBench*, a FHIR-compliant virtual EHR environment with multi-step, clinician-authored tasks spanning diverse categories (beyond QA). Our work builds directly on FHIR-AgentBench (Lee et al., 2025), which grounds thousands of clinician-sourced questions in real MIMIC-IV-FHIR records and systematically evaluates single-turn vs. multi-turn agents, natural-language vs. code-based reasoning, and different retrieval interfaces. We adopt their benchmark as our starting point.

## 2.3. Tool-Calling LLM Agents

Tool-calling LLM agents are commonly evaluated not only on whether they select the correct tool, but also on whether they emit executable calls and recover from downstream errors. Recent benchmarks emphasize multi-turn and stateful function-calling settings (Patil et al., 2025), while stable benchmarking efforts address the practical instability of real-world APIs and evaluation randomness (Guo et al., 2024). A consistent finding is that tool use is fragile: small deviations in argument values or formatting can trigger tool failures that propagate through a toolchain, even when the high-level intent is correct (Xiong et al., 2025). This fragility is amplified in multilingual interactions, where models may generate semantically correct parameter values in the user's language that violate execution conventions (Luo et al., 2026). Importantly, the *semantic specification* of tools (names, signatures, and natural-language descriptions) shapes the effective action space and reduces ambiguity in tool selection and argument construction, making it a key determinant of reliable tool use (Müller, 2025).

These failure modes are directly relevant to EHR backends such as FHIR, where strict schemas and traversal constraints make tool-call validity a first-order concern. A single malformed search parameter or an incorrectly typed reference field can silently return empty results, derailing an otherwise correct reasoning chain.

## 2.4. Reinforcement Learning for Tool Use

Privacy and data-security constraints in clinical settings often motivate on-premise deployment with open-weight models, which typically underperform closed models on realistic EHR tool-use benchmarks without specialization (Lee et al., 2025). Reinforcement learning from verifiable feedback/rewards (RLVF/RLVR) offers a practical route to close this gap. Group Relative Policy Optimization (GRPO), introduced in DeepSeekMath and adopted in DeepSeek-R1 (Shao et al., 2024; Guo et al., 2025), demonstrated that strong reasoning can emerge from automatically checkable signals, initially in mathematical domains (Guo et al., 2025). Follow-up analyses further study when and why such RLVR-style training succeeds (Liu et al., 2025b). The same principle extends to tool use, where execution outcomes provide a natural verifiable reward: recent work applies GRPO-style post-training to strategic tool integration, execution-grounded optimization, and tool-centric agentic reasoning (Feng et al., 2025; Singh et al., 2025; Zhang et al., 2025; Qian et al., 2025).

More recent work extends this line along several axes. On the *training* side, TAPO optimizes adaptive tool-calling policies that interleave reasoning with on-demand invocation of search and code tools (Wu et al., 2025); Fission-GRPO converts execution errors into on-policy corrective supervision rather than treating them as sparse negative rewards (Zhang et al., 2026); and ToolOrchestra trains compact 8B-parameter orchestrators to allocate calls across heteroge-

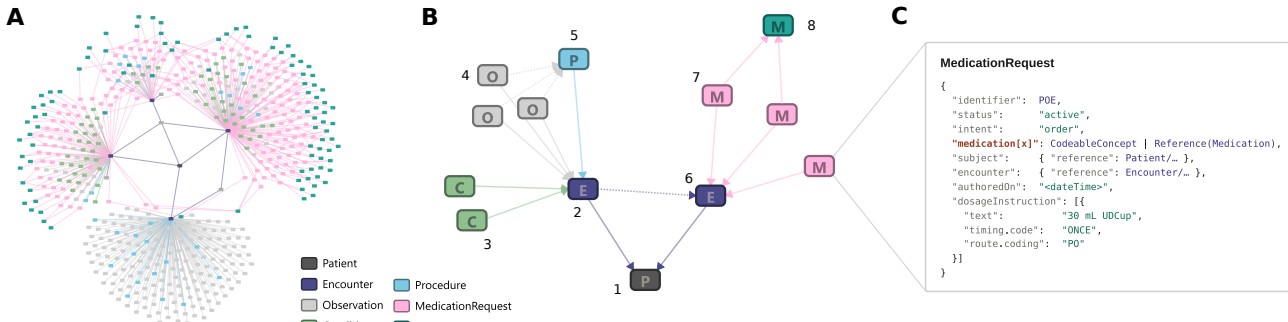

*Figure 2.* Fast Healthcare Interoperability Resources (FHIR). **(A)** Resource graph of a MIMIC-FHIR patient, connected by references. **(B)** Clinical concepts can be mapped to FHIR: A patient (1) presents to the emergency department (2; first Encounter) with one or more conditions (3). During the visit, clinical observations (4) were recorded (e.g. heart rate, blood pressure), each linked to the procedure that generated them (5, e.g. vital signs measurement). The patient was subsequently admitted to the hospital (6; second Encounter). During the inpatient stay, Furosemide (h) was requested at a dose of 40 mg (g). Notice that one of the three MedicationRequest resources does not reference a Medication resource. Instead, the medication information is contained in the MedicationResource itself (contained resource).

neous models and tools under outcome-, efficiency-, and preference-aware objectives (Su et al., 2025). Orthogonally, reliability can be improved without additional training by constraining tool interaction through Hoare-style contracts that gate invocation on verified preconditions and commit results only after runtime postcondition checks (Liu et al., 2026). On the *data* side, complementary pipelines bootstrap tool learning by synthesizing verifiable hard cases from agent failure graphs (Hao et al., 2026), converting exploratory trajectories into tasks that stress-test ambiguous and evolving user intents (Wang et al., 2026), and deriving virtual training trajectories from real tool calls with mutation-based negative samples to reduce intent deviation (Xiong et al., 2026).

In structured EHR backends such as FHIR, answers can be verified against the ground-truth record, making the setting a natural fit for RLVR-style post-training.

### 2.5. Interpreter-Mediated Reasoning

A line of work frames agent actions as executable programs rather than atomic tool calls, delegating precise computation to an external interpreter. PAL (Gao et al., 2023) and Program-of-Thoughts (Chen et al., 2022) generate code to separate natural-language reasoning from arithmetic, while ReAct (Yao et al., 2023) popularizes the general interleaving of reasoning and acting with environment feedback. Code-Act (Wang et al., 2024) instantiates this pattern with Python code as action. SkyRL-SQL (Liu et al., 2025a) demonstrates that such agents benefit from RL post-training with execution-grounded rewards in multi-turn Text-to-SQL; our setting is analogous but structurally different, applying RL post-training to CodeAct agents over schema-constrained FHIR graphs.

## 3. Method

### 3.1. Problem Definition

A patient's record in FHIR forms a typed, directed, heterogeneous graph (§2.1). Answering a clinical question over a FHIR graph requires selecting resource types, traversing references across hops, and applying client-side computation (filtering, temporal alignment, aggregation, etc.). We focus on the *reading* side of interoperability: given access to an existing FHIR server, how can an agent reliably retrieve and reason over the information needed for a correct answer?

The central difficulty is that FHIR standardizes interfaces but not content: local profiles, extensions, and terminology choices mean the same clinical concept can appear in different resources or fields, and optional elements are populated inconsistently across sites (Kramer, 2023). A single-shot query plan that assumes a fixed schema is therefore brittle. We instead adopt an iterative pattern, *schema discovery*, in which the agent probes candidate resources, inspects returned JSON, and refines its strategy before committing to an answer.

### 3.2. System Architecture

We implement post-training with SKYRL (Griggs et al., 2025), which provides an end-to-end RL pipeline comprising (i) a trainer (we use GRPO-style updates; §3.3), (ii) a high-throughput rollout engine based on vLLM, and (iii) an environment that defines tool interfaces, parses actions, and returns observations.

Episodes follow a ReAct-style loop: after each observation (tool output or error message), the agent produces a brief reasoning step and then either issues a tool call or outputs a final answer. We cap each episode at $N_{\max} = 6$ turns,

where a turn is either a single tool invocation or a final answer.

**Environment and tools.** The environment exposes two tools and a terminal action:

1. A *Retrieval* tool takes a patient ID and a resource type and queries all resources of that type for the given patient. For resource types not directly linked to *Patient* (e.g., *Medication*), the tool first fetches patient-linked *MedicationRequest* resources and then resolves the referenced *Medication* entries. The results are stored in a shared workspace, accessible for the Python tool.

2. A *Python* tool executes Python code over the retrieved resources, enabling both schema inspection (by printing samples into context) and arbitrary data manipulation (filtering, sorting, temporal alignment, unit conversions, aggregation, etc.).

3. Finally, a *Finish* action carries the agent's final answer and terminates the episode. It is exposed as a tool for interface uniformity but does not interact with the environment.

The retrieval tool is inspired by Lee et al. (2025); we intentionally prioritize simplicity over efficiency and leave optimized query planning to future work. Actions are emitted as XML-tagged tool calls in the native Qwen3 format and executed by the environment; for non-terminal actions, the resulting output is fed back as the next observation.

**FHIR server.** We deploy a BLAZE FHIR server (The Samply Community, 2025) and populate it with the 100 MIMIC-FHIR demo patients used by FHIR-AGENTBENCH. If not stated otherwise, results are reported on the 424 FHIR-AGENTBENCH validation set questions.

**Prompting.** The system prompt was intentionally kept short:

```
You are a FHIR data analyst.  Answer
patient data questions by querying a
FHIR server.  Rules:
  • Every claim must trace to a
    print() output or computation.
  • If unsure about a resource's
    schema, print a sample first.
  • Keep your reasoning brief.
  • When done, call finish.
```

The prompt is followed by tool descriptions (Appendix A.1), which are serialized into the system prompt by the Qwen3 chat template.

### 3.3. Training

**Formulation.** Each episode begins with a clinical question $q$ and an execution context $c$ (patient identifier, time horizon, task-specific constraints). At step $t$ the agent observes the question and the full interaction history $h_t = (q, c, a_1, o_1, t_1, \ldots, a_{t-1}, o_{t-1}, t_{t-1})$ and selects an action $a_t$, i.e. a code snippet executed by the runtime (§3.2). The runtime returns observation $o_t$ (retrieved resource counts, printed information, or error messages), which is appended to the history. Each $t_i$ denotes an optional thinking block produced by the agent after observation $o_i$. An initial thinking step $t_0$ may optionally precede the first action. This multi-turn structure is what enables schema discovery: early actions probe candidate resources and inspect how information is recorded, while later actions exploit those findings to retrieve and aggregate the answer.

A trajectory has the form $\tau = (q, c, a_1, o_1, t_1, \ldots, a_T, o_T, t_T, y)$, where $y$ is the final natural-language answer emitted at step $T$. An episode terminates when the agent produces $y$ or exceeds the turn budget $N_{\max}$ or token budget $L_{\max}$. We optimize expected answer correctness:

$$\max_{\theta} \ \mathbb{E}_{(q,c)\sim\mathcal{D}, \ \tau\sim\pi_\theta(\cdot|q,c)}\big[\, r(\tau) \,\big], \qquad (1)$$

where $\pi_\theta$ is the agent policy (an LLM parameterized by $\theta$) and $r(\tau) \in \{0, 1\}$ indicates whether $y$ is correct.

**LLM judge.** We use Qwen2.5-72B-Instruct as an automatic judge, following the evaluation protocol and prompting scheme of Lee et al. (2025). The trajectory receives $r(\tau) = 1$ if the output matches the required answer format **and** the predicted answer is judged correct; otherwise $r(\tau) = 0$.

**GRPO.** We optimize Eq. 1 with Group Relative Policy Optimization (GRPO; Shao et al., 2024), which eliminates the need for a learned value function by estimating advantages from a group of sampled trajectories. For each query $q$ the current policy $\pi_{\theta_{\text{old}}}$ samples a group of $G$ trajectories $\{\tau_i\}_{i=1}^{G}$ with corresponding binary rewards $\{r_i\}_{i=1}^{G}$. The advantage of the $i$-th trajectory is:

$$\hat{A}_i = \frac{r_i - \text{mean}(\{r_j\}_{j=1}^{G})}{\text{std}(\{r_j\}_{j=1}^{G})}. \qquad (2)$$

GRPO maximizes a clipped surrogate with a KL penalty toward a reference policy $\pi_{\text{ref}}$ which we update every epoch

with the latest policy weights:

$$J_{\text{GRPO}}(\theta) = \mathbb{E}_{q \sim \mathcal{D},\ \{\tau_i\} \sim \pi_{\theta_{\text{old}}}(\cdot | q)} \left[ \frac{1}{G} \sum_{i=1}^{G} \right.$$

$$\frac{1}{|\tau_i|} \sum_{t=1}^{|\tau_i|} \Big( \min\big(\rho_{i,t}(\theta)\, \hat{A}_i, \tag{3}$$

$$\text{clip}\big(\rho_{i,t}(\theta),\, 1 - \epsilon_{\text{low}},\, 1 + \epsilon_{\text{high}}\big)\, \hat{A}_i\big)$$

$$\left. - \beta\, D_{\text{KL}}\big(\pi_\theta \,\|\, \pi_{\text{ref}}\big) \Big) \right],$$

with importance ratio $\rho_{i,t}(\theta) = \pi_\theta(\tau_{i,t} \mid q, \tau_{i,<t}) / \pi_{\theta_{\text{old}}}(\tau_{i,t} \mid q, \tau_{i,<t})$, where $\tau_{i,t}$ denotes the $t$-th agent-generated token in trajectory $\tau_i$.

Each completion $\tau_i$ in our setting spans multiple interaction turns: it comprises all agent-generated tokens (code snippets and the final answer) across the episode; environment observations are excluded from the likelihood. The reward $r_i$ is assigned once at episode end, so the per-token advantage $\hat{A}_i$ is constant across all tokens in trajectory $i$. During training, multiple queries form a batch and gradients are averaged.

# 4. Experiments

We use instruction-tuned Qwen3 models with native tool calling and reasoning.

## 4.1. Baselines

FHIR-AgentBench reports an answer correctness of 50% with o4-mini in a multi-turn CodeAct setting (Lee et al., 2025). In our harness, o4-mini achieves a comparable 47%. We additionally include Gemini-3-Flash (52%) and GLM-5 (59%, 744B parameters with 40B active) as API-based baselines. To isolate the contribution of RL training (see Section 4.2), we evaluate Qwen3 across four model sizes (4B, 8B, 14B, 32B) in a zero-shot setting. For each prompt, we perform five inference passes per prompt (5×424 rollouts total; temperature 0.1, 12-turn budget), reporting the mean score, standard deviation, and pass@5 in Table 1.

## 4.2. RL Training

Qwen3-8B was trained with a fixed learning rate of $1 \times 10^{-6}$ and a batch size of 8. We focus on the 8B model here; results for applying the best recipe to the 1.7B and 4B models are reported in Section A.3. For variance reduction, we use a group size of 8 (i.e., eight rollouts per prompt) and restrict to a 12,000-token context length. Train rollouts were sampled with a temperature of 1.0, whereas a temperature of 0.1 was used for evaluation rollouts. We ran three RL recipes:

1. GRPO baseline with symmetric clipping ($\epsilon_{\text{low}} =$

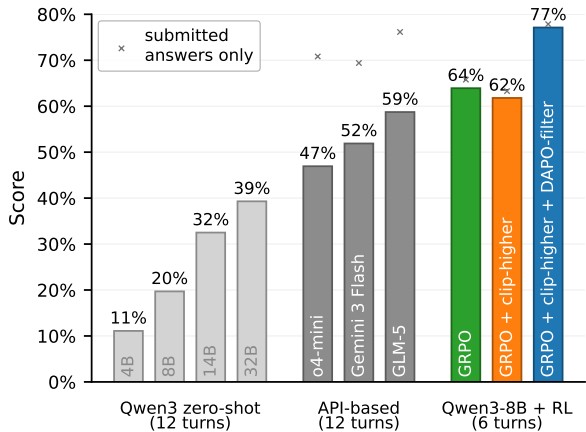

*Figure 3.* Answer correctness on FHIR-AgentBench (Lee et al., 2025) of vanilla Qwen3 models in different sizes (4B to 32B), API-based models (closed/open-weights), and after RL training (step 3000). Small crosses indicate performance restricted to questions where the agent successfully submitted an answer.

$\epsilon_{\text{high}} = 0.2$) and token-mean normalization.

2. Like 1., but with asymmetric clipping ("clip-higher"; $\epsilon_{\text{low}} = 0.2$, $\epsilon_{\text{high}} = 0.28$) instead of symmetric clip.

3. Like 2., but with dynamic sampling as proposed by the DAPO paper (Yu et al., 2025): zero-variance groups (all correct or all incorrect) are filtered out from training.

Symmetric and asymmetric clipping show nearly identical performance after 3000 training steps. Answer correctness reaches 64% for setup 1 and 62% for setup 2 (Figure 4). Given this negligible difference, we introduce dynamic sampling as a third recipe. Setup 3 reaches 77% correctness in fewer steps, suggesting that filtering zero-variance groups improves sample efficiency. We additionally explored an SFT warm-start as an alternative route to faster convergence, but found it did not improve over RL alone (Section A.2). We observe that RL training reduces the number of turns needed to submit a solution, i.e., the model learns to answer within a six-turn budget (Figure 5). This shows a clear effect of RL training beyond mere elicitation: answer correctness increases while the number of turns required decreases.

## 4.3. Judge

We measured agreement between the automated judge (Qwen2.5-72B-Instruct) and human annotations from a board-certified physician. Annotations cover the 424 validation samples from the base-GRPO run (setup 1) at step 3000. The resulting agreement is 96.2% (precision = 96.7%, recall = 97.4%), in line with the 97% reported by Lee et al. (2025).

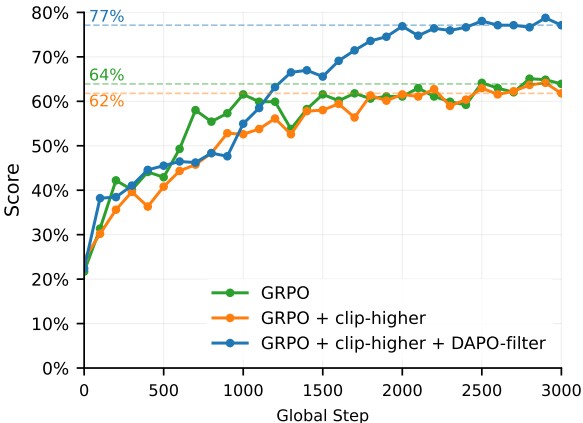

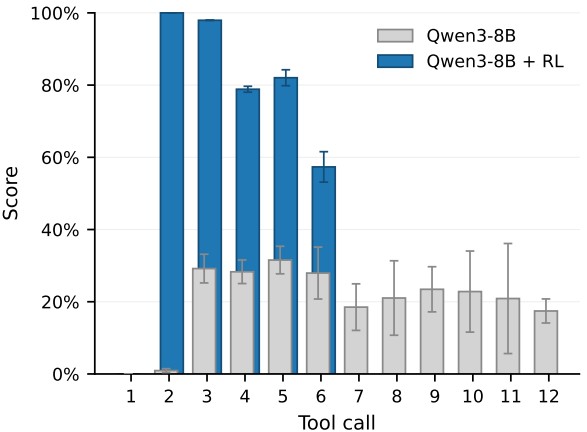

*Figure 4.* Training curves for Qwen3-8B. Curves show the answer correctness on the FHIR-AgentBench (Lee et al., 2025) validation subset (n=424) with a 6 turn budget. Recipes: vanilla GRPO (blue), +clip-higher (orange), +DAPO-filter (green). Dashed horizontal lines mark each run's score at step 3000.

*Figure 5.* Per-turn score for vanilla Qwen3-8B (12-turn budget, grey) vs. GRPO-trained Qwen3-8B at step 3000 (6-turn budget, blue). Error bars (±1 SD) capture variability across rollouts within each bin, with five rollouts per prompt.

*Table 1.* Answer correctness on FHIR-AgentBench (%). Mean ± SD computed over 5 rollouts for Qwen3 models.

| MODEL | SCORE | PASS@5 |
|---|---|---|
| QWEN3-4B | $11.1 \pm 1.3$ | 29.5 |
| QWEN3-8B | $19.7 \pm 0.3$ | 47.2 |
| QWEN3-14B | $32.5 \pm 1.0$ | 53.8 |
| QWEN3-32B | $39.3 \pm 1.3$ | 68.6 |
| O4-MINI | 46.9 | – |
| GEMINI 3 FLASH | 51.9 | – |
| GLM-5 | 58.7 | – |
| QWEN3-8B + RL[†] | $76.7 \pm 0.9$ | 79.7 |

[†]GRPO + clip-higher + DAPO-filter

For the API-based baselines, we use the same Qwen2.5-72B-Instruct model served via the OpenRouter API in fp8 precision. Agreement between the OpenRouter judge and our local vLLM judge (fp16) was measured at 99.75%, confirming that the quantization difference has negligible impact.

### 4.4. Breakdown

To analyze failure modes, we categorize validation questions by the ground-truth FHIR resource type required to answer them, which is annotated in the dataset (Figure 6). Among the 424 questions in the validation split, most require fetching Encounter ($n = 64$), Observation ($n = 155$), Medication/MedicationRequest ($n = 71$), or belong to the Empty category ($n = 110$).

All questions require filtering resources of the relevant type by patient as a first step, using the query tool. For Encounter questions, the agent performs Python post-processing such

as selecting relevant fields or sorting by date. Observation questions additionally require string-matching the *code.coding[0].display* field to locate a specific lab value (or other values), followed by filtering analogous to the Encounter case. Medication questions require resolving the MedicationRequest-Medication reference: the agent must fetch both resource types and surface the Medication ID from MedicationRequest into context in order to dereference it (see Figure 11). The Empty category contains questions whose ground-truth FHIR ID list is empty, meaning the agent must conclude that no matching record exists; reward is given only if the agent reports a negative or zero answer.

The agent performs worse on questions that require following the Medication reference. While the agent succeeds on some prompts (Figure 11), it does not apply this skill reliably. We hypothesize that longer training would improve performance further: the skill has been acquired but is applied inconsistently.

For the best RL run (setup 3: GRPO + clip-higher + DAPO-filter), we also show the resource-type breakdown over training steps (Figure 7). The agent first learns to commit to a no-answer, which is the easiest reward signal to get. Eventually, it learns to answer questions that require interacting with the data, with some negative transfer to the Empty category. Overall performance improves with training, though spikes in one curve often coincide with drops in others.

## 5. Conclusions

Privacy constraints often make on-premise deployment a necessity for clinical AI; however, smaller open-weight models typically perform worse on realistic EHR tasks than

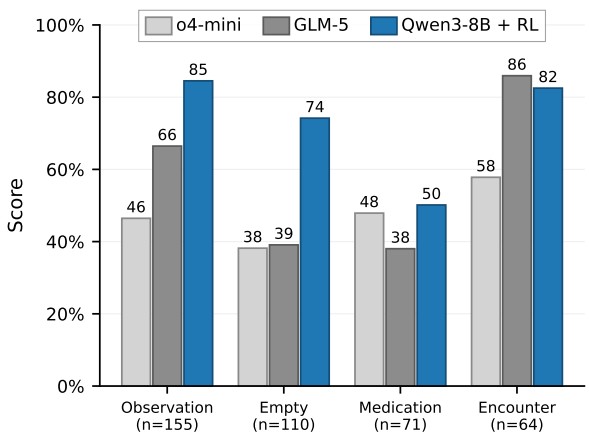

*Figure 6.* Answer correctness by FHIR resource type. Empty refers to questions with no matching FHIR resources (i.e., negative or null answers). Categories with fewer than 15 validation samples (Location, Procedure, Patient, Condition) are omitted.

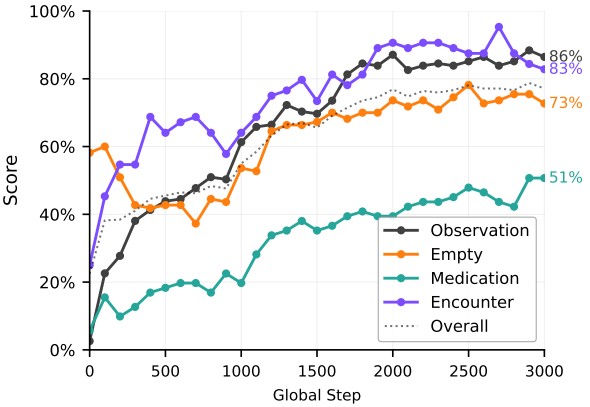

*Figure 7.* Our best RL run (GRPO + clip-higher + DAPO filter), broken down by FHIR resource type over training steps. The model first learns to return "Empty" on questions with no matching resources, which is the cheapest reward to get, before improving on the other resource types. The dotted line shows the overall mean.

their closed-source counterparts. We show that with the right post-training recipe, a small open model can exceed closed-model performance on structured clinical reasoning.

We frame FHIR question answering as a sequential decision-making problem over a typed, heterogeneous resource graph, and instantiate it as a multi-turn CodeAct agent that probes schema variants, retrieves resources, and processes JSON in Python. The agent is post-trained with GRPO using execution-grounded rewards from an LLM-judge, validated at 96% agreement with a board-certified physician.

The results suggest that the limiting factor for open models on FHIR was not capacity but specialization: the trained agent learns the conventions of the underlying server through repeated interaction and debugging, rather than relying on an idealized schema.

Inspection of trajectories indicates that most remaining failures involve reference traversal across resource types, e.g., resolving MedicationRequest-Medication references, which the agent does not yet do consistently. Future FHIR benchmarks could target this capability directly by including harder questions that require traversing three or more resource types.

Several limitations qualify this picture. Our retrieval interface fetches whole resource types and will not scale efficiently to large patient populations without a planning layer for targeted FHIR search. Evaluation depends on an LLM judge. While agreement with our physician annotator is high (96%), the remaining disagreement bounds how precisely further improvements can be measured. Finally, while 77% answer correctness is a substantial gain, it remains well below the reliability bar required for high-stakes clinical use.

Future work should test whether the recipe transfers to larger backbones (e.g., 32B) and replace the simple retrieval tool with learned query planning to scale to larger populations.

More broadly, our findings indicate that execution-grounded RL post-training is a viable path to capable on-premise clinical agents, closing the open-versus-closed gap in exactly the setting where on-premise deployment is most needed.

## Impact statement

This work contributes to improving the robustness of LLM-agents for structured clinical data reasoning, with potential to support more scalable and interoperable healthcare analytics. By showing that a small open-weight model can match or exceed closed commercial models, our approach enables fully on-premise deployment, allowing institutions to reason over sensitive patient records without transmitting data to third-party APIs. This privacy-preserving capability is a primary motivation for our setting and a direct benefit of the recipe we present. These benefits come with real risks. In clinical settings, agent errors carry concrete consequences: a missed medication or misattributed lab value could mislead downstream decisions. Our best system reaches 77% answer correctness, well below the bar for autonomous clinical use. Furthermore, our results are obtained on a single de-identified dataset (MIMIC-IV) and a single FHIR server; because FHIR deployments vary widely in profiles, extensions, and terminology, performance and failure modes may differ at other sites, and the system should be re-validated before use in any new environment. We therefore view this work as a step toward decision-support tools that require human verification, not replacement of clinical judgment.

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

# A. Appendix

## A.1. System prompt

The system prompts for all experiments consist of two parts. First, the natural-language instruction:

```
System prompt pt. I

You are a FHIR data analyst. Answer patient data questions by querying a FHIR server.
Rules:
- Every claim must trace to a print() output or computation.
- If unsure about a resource's schema, print a sample first.
- Keep your reasoning brief.
- When done, call finish.
```

Second, the tool schemas. Since we use native tool calling, every chat template serializes the tool schema differently. The Qwen3 chat template serializes the tool schemas into the following string:

---

**System prompt pt. II**

```
# Tools
You may call one or more functions to assist with the user query.
You are provided with function signatures within <tools></tools> XML tags:
<tools>
{"type": "function",
 "function": {
   "name": "fhir_query",
   "description": "Query a FHIR server for health records. Retrieved resources are
   accumulated across calls. Use multiple calls to gather all the data you need before
   answering.",
   "parameters": {
     "type": "object",
     "properties": {
       "resource_type": {"type": "string",
         "description": "FHIR resource type to query, e.g. Patient, Condition,
         Observation, MedicationRequest, Procedure, etc."},
       "patient_fhir_id": {"type": "string",
         "description": "Patient FHIR ID to filter by."}},
     "required": ["resource_type", "patient_fhir_id"]}}
}
{"type": "function",
 "function": {
   "name": "python",
   "description": "Execute Python to analyze or transform FHIR
   data.\n`retrieved_resources` (dict of resource_type -> list[dict]) only
   contains\ndata from prior fhir_query calls – fetch before you analyze.\nOnly
   printed output is returned.",
   "parameters": {
     "type": "object",
     "properties": {
       "code": {"type": "string",
         "description": "Python code to execute. Use print() to produce output."}},
     "required": ["code"]}}
}
{"type": "function",
 "function": {
   "name": "finish",
   "description": "Signals the completion of the current task or conversation.\n\nUse
   this tool when:\n- You have successfully completed the requested task\n- You cannot
   proceed further due to technical limitations or missing information\n\nThe answer
   field should include the final answer to the problem (follow the required format)
   if an answer is required by the problem.\n",
   "parameters": {
     "type": "object",
     "properties": {
       "answer": {"type": "string",
         "description": "Final message summarizing the task or containing the
         answer."}},
     "required": ["answer"]}}
}
</tools>
For each function call, return a json object with function name and arguments within
<tool_call></tool_call> XML tags:
<tool_call>
{"name": <function-name>, "arguments": <args-json-object>}
</tool_call>
```

## A.2. SFT as RL Warm-Start

We investigate supervised fine-tuning (SFT) as a warm-start for RL training, hypothesizing that it would reduce overall training time.

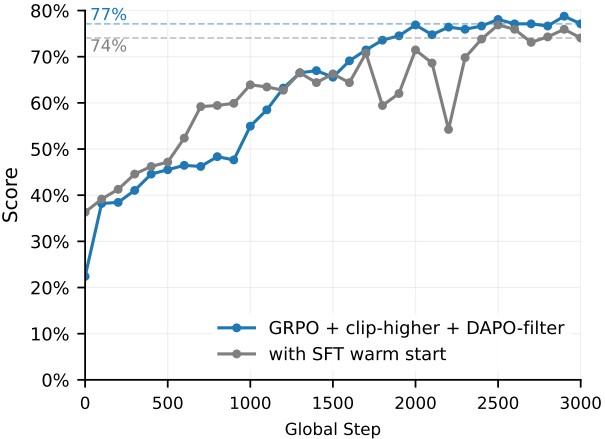

*Figure 8.* RL warm-start with SFT.

**Trajectory collection.** Initial experiments using GLM-5 trajectories were unsuccessful: Qwen3-8B could not reliably execute the longer and more complex Python code produced by GLM-5, introducing syntax errors from which it could not recover within the 6-turn budget. We therefore collected trajectories from the same model family (Qwen3-32B) to remain closer to on-policy. Using temperature $0.7$ and 12 rollouts per prompt over the training set ($n = 2098$), we obtained 25,176 trajectories. To address weak performance on medication-related questions observed in earlier runs, we extended the system prompt with a hint on resolving the `MedicationRequest-Medication` reference (see Section 4.3), yielding a pass@12 of $0.78$ with a 12-turn budget.

**Filtering.** We discarded incorrect trajectories (16,176 removed) and capped tool-call counts at 6, except for medication questions which retained the 12-turn budget. This yielded 6,264 eligible trajectories spanning 1,447 unique questions. We deduplicated by selecting the shortest trajectory (fewest tool calls) per question, resulting in a final SFT set of 1,447 rollouts.

**Training setup.** We supervise-fine-tune Qwen3-8B with assistant-only loss using LoRA applied to all linear projections (rank 16, $\alpha = 32$, dropout 0.05), optimized with 8-bit AdamW at a peak learning rate of $2 \times 10^{-5}$. We train for one epoch with an effective batch size of 16.

**Results.** Zero-shot answer correctness improves from $20\%$ to $36\%$ after SFT, validating the approach. We then apply RL training (recipe 3: GRPO + clip-higher + DAPO-filter) starting from the SFT checkpoint. While the initial score is higher than the corresponding RL-only run, training is unstable and eventually collapses, peaking at $76\%$ answer correctness; below the $\sim 80\%$ reached by recipe 3 without SFT warm-start (Figure 8).

### A.3. Model Size Ablation

In addition to the 8B Qwen3 model, we also experiment with smaller model sizes during RL training: the 4B and 1.7B Qwen3 models. For all runs we use setup 3 (GRPO + clip-higher + DAPO-filter), as it performed best for the 8B model. See Figure 9 for training curves.

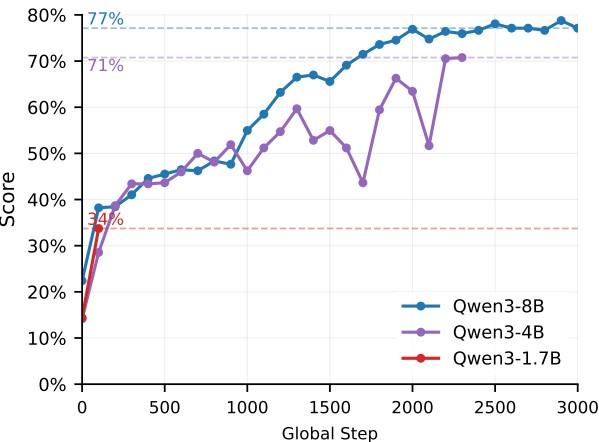

*Figure 9.* Comparison of three model sizes trained with recipe 3 (GRPO + clip-higher + DAPO-filter). The 4B and 1.7B runs did not reach step 3000: at a certain point, no groups with variance greater than zero remained, so the DAPO filter discarded all samples and training stopped.

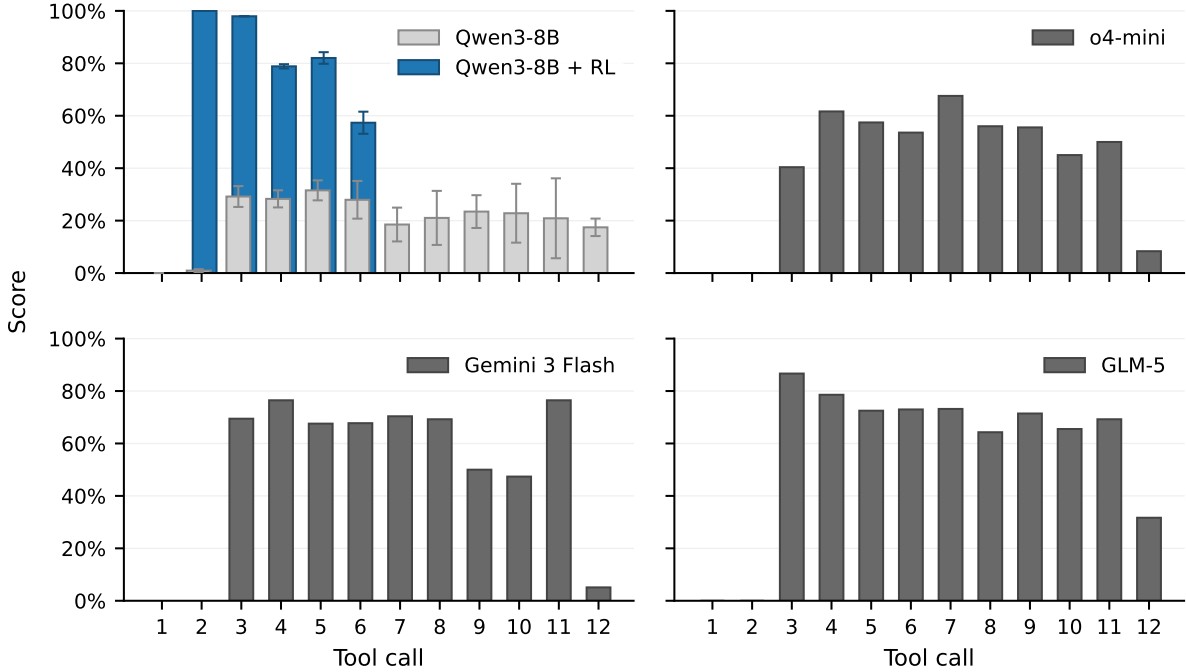

*Figure 10.* Accuracy by FHIR resource type for Qwen3 (zero-shot and trained), and API-baselines.

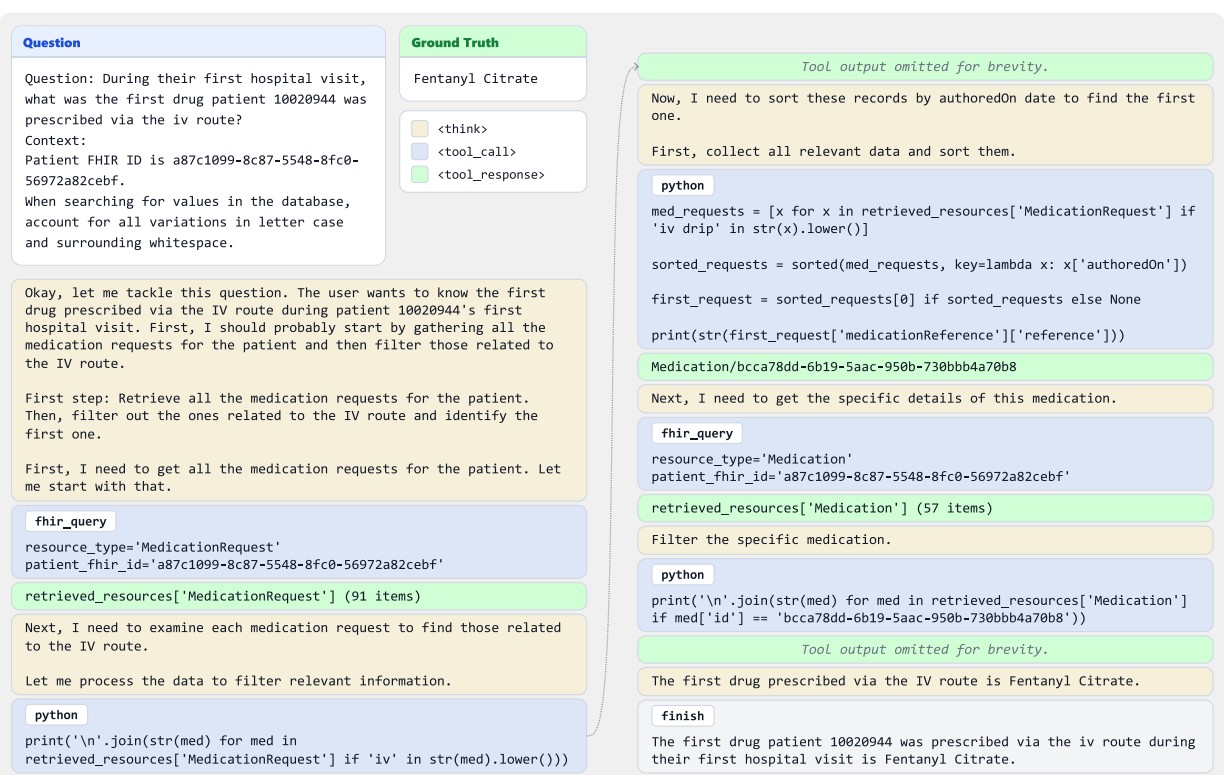

*Figure 11.* Example trajectory. The agent first retrieves all MedicationRequest resources. Then it filters for "iv", and sorts the resources. It identifies the correct MedicationRequest resource, and then resolves the reference to Medication. Medication holds the actual medication that the question is looking for. In this plot, Python tool outputs were collapsed for brevity.

