# OpenReview forum: "Reinforcement Learning for Tool-Calling Agents in Fast Healthcare Interoperability Resources (FHIR)"
_ICML.cc/2026/Conference — ICML 2026 regular_

### Official Review · Reviewer_CC6T · 2026-03-07

**Soundness:** 2
**Presentation:** 2
**Significance:** 2
**Originality:** 2
**Overall Recommendation:** 4
**Confidence:** 3

**Summary:**

This paper frames clinical data retrieval from FHIR servers as a sequential decision-making problem and applies reinforcement learning to train tool-calling agents. The authors use a CodeAct-style agent architecture where the model generates interleaved thought, tool-call, and Python code actions to query FHIR resources and answer clinical questions. Two experiments are conducted on FHIR-AgentBench (424 questions from synthetic FHIR data). The paper also provides a trajectory viewer tool for inspecting agent behavior.

**Compliance With Llm Reviewing Policy:**

Affirmed.

**Final Justification:**

My previous concerns have been solved and I would like to keep my current assessment.

**Key Questions For Authors:**

see above

**Limitations:**

yes

**Strengths And Weaknesses:**

Strength: FHIR is the dominant interoperability standard in healthcare IT, yet querying FHIR data programmatically requires understanding complex resource relationships, reference traversal, and data normalization. Framing this as an RL problem for tool-calling agents is novel and addresses a real need: clinicians and data analysts spend significant time writing FHIR queries. The CodeAct agent design with think-tool-observation loops is well-suited to the multi-hop reasoning required for FHIR queries.

====

Weakness:

W1. Experiment 1 (original prompt + RL) achieves 55%, while Experiment 2 (improved prompt + RL) achieves 64%. The improved prompt includes substantial domain-specific heuristics (print data before analyzing, normalize strings, handle None, evidence required). The paper does not report what the improved prompt achieves WITHOUT RL training (i.e., zero-shot with the improved prompt on Qwen2.5-Coder-7B) (maybe I missed but please identify). Without this ablation, it is impossible to determine how much of the Exp. 2 improvement comes from RL versus prompt engineering. Given that prompt engineering alone often yields substantial gains on tool-use tasks, this confound seriously undermines the claimed RL contribution.

W2. The paper compares only against o4-mini and a few models as baselines. Some missings are: (a) comparisons with other open-source coding models of similar scale (e.g., DeepSeek-Coder, CodeLlama); (b) comparisons with SFT-only training on the same trajectories (to isolate RL's marginal contribution over behavioral cloning); (c) comparisons with other RL algorithms (PPO, DPO, REINFORCE) to justify Dr. GRPO specifically; (d) evaluation of larger models (e.g., 32B, 70B) to understand the accuracy-scale relationship (if resources allowed, optional). The claim that RL training is essential for this task is not adequately supported without these ablations.

W3. The training and evaluation use the same synthetic FHIR data distribution (Synthea). Real-world FHIR servers differ dramatically: they contain vendor-specific extensions, inconsistent coding systems, and data quality issues. The paper does not evaluate on any real-world FHIR data, nor does it discuss how the agent would handle these practical challenges.

---

> ### Author Rebuttal · Authors · 2026-03-31
>
> Thank you reviewer CC6T07 for the feedback and the questions.
>
> Since there was a consensus between the reviewer critique, we have addressed the paper's major weaknesses in the response to reviewer yPmE in more detail.
>
>
> W1: to better separate RL-contribution from our model harness (especially prompting), we scored the “vanilla” Qwen performance (around  23.6% answer correctness, max 12 turns) on top of the RL-trained Qwen performance (64%). We also re-ran the experiment in a more controlled setup with a shared set of “base hyperparameters" (learning rate fixed to 1e-6, 8 rollouts, bs 8), where we only change one parameter at a time. Surprisingly, a smaller 4B model performed almost equally well as the 8B model.
>
> W2: We have included more closed source baselines (o4-mini and gemini3 flash with our harness), which both score around 50%
>
> W3: We thank the reviewer for this important point. We want to clarify a potential misconception: we use the MIMIC-FHIR dataset, which is derived from real clinical records (MIMIC-IV), not Synthea-generated synthetic data. This distinction matters because MIMIC-FHIR reflects genuine clinical documentation patterns rather than idealized synthetic ones.
> That said, we fully acknowledge that the reviewer's core concern stands: our training and evaluation occur on a single FHIR deployment with a fixed profiling convention, and real-world FHIR servers introduce additional challenges such as vendor-specific extensions, inconsistent coding systems, and variable data quality that are not represented in our benchmark. We do observe multi-turn self-correction behavior in our agent (e.g., recovering from malformed Python or unexpected API responses), which provides a basis for adapting to schema variation at inference time. However, we agree that this does not constitute a systematic evaluation of robustness.
> Constructing a diverse multi-site FHIR evaluation set is a significant independent effort that falls outside the scope of this paper, which we position primarily as a framework and training infrastructure contribution. We explicitly flag cross-site generalization as an important direction for future work and made this limitation more prominent in the revised manuscript.
>
> #####
> Ongoing rebuttal flow:
> #4 - More / better baselines to understand the contribution of RL
> We streamlined our experimental setting. We begin by a baseline of Qwen3-8B using our harness. While we train Qwen3-8B to perform the task in up to 6 steps we evaluate the base model in addition with a budget of 12 steps.
> We have therefore added / are adding the following comparisons (using the:
> Base Qwen3 8B in our harness, with [same number of tool-interaction rounds / same rollout budget]: 21.7%
> Base Qwen3 8B in our harness, no RL, with [12  tool-interaction rounds, so twice of what we RL to]: 23.6%
> GRPO-trained Qwen3 8B: 64%
> GRPO with asymmetric clipping Qwen3 8B: 63%
> These results let us separate different effects that were less clearly disentangled in the original submission:
> the effect of moving from a generic model setup to a domain-specific tool-using harness
> the effect of the training algorithm itself
> Our expectation is that this will make the paper’s empirical story much sharper: the harness matters, but RL contributes additional gains beyond simply wrapping the model in the environment.

---

> > ### Author Rebuttal · Reviewer_CC6T · 2026-04-02
> >
> > W2 is not addressed. The results look good.

---

> > > ### Author Response · Authors · 2026-04-07
> > >
> > > Thank you for the discussion and follow-up reviewer CC6T.
> > >
> > > We have expanded the comparison set in the direction you asked for.
> > >
> > > Concretely, we now report
> > > - base Qwen3-8B in the same harness with the matched interaction budget (6 budget) without RL: 21.7%
> > > - base Qwen3-8B in the same harness with the matched interaction budget (12 budget) without RL: 23.6%
> > > - GRPO-trained Qwen3-8B (6 turns budget):  64%
> > > This separates the effect of the harness from the effect of RL.
> > >
> > > In addition, we are adding two further ablations: a same-scale open-source coder baseline, and SFT-only / SFT+RL comparisons. These are the key missing comparisons for testing whether the gain is specifically from RL rather than from stronger scaffolding or imitation alone. We will update the paper text accordingly

---

### Official Review · Reviewer_8uzj · 2026-03-13

**Soundness:** 2
**Presentation:** 2
**Significance:** 3
**Originality:** 2
**Overall Recommendation:** 4
**Confidence:** 4

**Summary:**

This paper addresses question answering over HL7 FHIR clinical data graphs by framing it as a sequential decision-making problem. The authors implement a multi-turn CodeAct agent and post-train a Qwen-2.5-7B model using GRPO-based reinforcement learning with execution-grounded rewards from an LLM judge. Evaluated on FHIR-AgentBench, the trained model improves answer correctness from 50% to 64%. The paper also provides a detailed analysis of failure modes broken down by FHIR resource type, showing that explicitly encouraging schema inspection during training substantially helps with Medication/MedicationRequest questions.

**Compliance With Llm Reviewing Policy:**

Affirmed.

**Final Justification:**

The additional experiments meaningfully strengthen the paper. Some concerns remain partially open. The confounded design between Experiments 1 and 2 is not ablated, and comparisons against supervised fine-tuning and other open-weight base models would further clarify the contribution.

**Key Questions For Authors:**

1. What is the performance of the base Qwen-7B model using the Experiment 2 system prompt? This is critical for understanding how much of the gain comes from prompt engineering vs. RL.
2. How sensitive are the results to the LLM judge? Have you measured judge accuracy against human annotations on a subset, and have you tried alternative judges?
3. The 8-turn budget seems potentially limiting for complex multi-hop questions. Did you experiment with different turn budgets, and how does performance vary?
4. Do you observe any reward hacking behaviors?

**Limitations:**

See Weaknesses and Key Questions.

**Strengths And Weaknesses:**

Strengths

Well-motivated problem setting. FHIR is genuinely the dominant interoperability standard, and the mismatch between its flexible schema and brittle LLM tool-use is a real practical pain point. The paper does a good job articulating why single-shot prompting fails and why iterative multi-turn interaction is a natural fit.

Clear end-to-end pipeline. The paper presents a complete recipe including environment construction, harness integration, reward design, and training, that is reproducible in principle.

Weaknesses

Limited experimental rigor and baselines. The paper compares against a single baseline o4-mini and two variants of the authors' own method. Several important comparisons are missing: 1) the base Qwen-7B model with the same prompts but no RL training, 2) supervised fine-tuning on expert trajectories as an alternative to RL, 3) other open-weight models of comparable size. Without these, it is difficult to disentangle the contributions of the prompt engineering, the CodeAct formulation, and the RL training itself.

Confounded experimental design between Exp. 1 and Exp. 2. Experiment 2 changes multiple variables simultaneously: the system prompt, the normalization scheme, and the learning rate. The paper attributes the gains to the combination but cannot identify which change matters most. An ablation study is needed.

Small evaluation scale and no test set. All results are on the 424-question validation set. No held-out test set results are reported, raising concerns about implicit overfitting through checkpoint selection and prompt tuning on the validation set. Additionally, some resource-type categories have very small n, making the per-category numbers unreliable.

---

> ### Author Rebuttal · Authors · 2026-03-31
>
> Thank you reviewer 8uzj13 for the feedback and the questions.
>
> We have addressed the paper's major weaknesses in the response to reviewer yPmE.
>
> Q1: this was addressed by running with the updated (simple prompt). The performance is around 23.6% (Qwen3-8B) before RL training (12 turn budget), and 64% after RL training (6 turn budget).
>
> Q2: this was also addressed (see response to reviewer yPmE).
>
> Q3: We agree that, in general, different questions may require different numbers of interaction steps. However, our turn-wise analysis suggests that the main limitation on the current benchmark is not simply the absolute turn budget. The detailed plot [https://limewire.com/d/f2JuG#QYyUqgHPeV
> ] shows two complementary patterns. First, for the models evaluated with a 12-turn budget (base Qwen3-8B, Gemini-3-Flash, and o4-mini), performance is concentrated in short-to-medium trajectories, while accuracy drops markedly for the longest trajectories. In particular, for Gemini-3-Flash and o4-mini, the high-count 11-turn bin has very low success, indicating that allowing more steps does not reliably rescue difficult instances. Second, our GRPO-trained Qwen3-8B is trained under a 6-turn cap, and thus explicitly optimized to solve tasks in few steps; correspondingly, its successful trajectories are concentrated in 2–5 turns. Taken together, these results suggest that, on this benchmark, the key issue is efficient use of a small number of steps, rather than access to a larger turn budget. We agree that extending the benchmark toward tasks that genuinely require systematically longer horizons is an important direction for future work.
>
>
> Q4: Thanks for the interesting question; yes we did. In our earlier manuscript version we relied on Qwen2.5-32B as the Judge. Occasionally, this led to reward hacking the 32B Judge:
> Model submitted a placeholder answer, such as "solution", "final answer", "answer".
> Model submitted code to (allegedly) compute the final answer instead of the real answer.
> Model restates or rephrases the question without giving an answer (e.g. "The maximum value of X for patient Y was:") without providing an actual answer.
> These were the main ways of reward hacking that we experienced. To prevent reward hacking, we switched to a 72B Qwen2.5 and did not experience any more reward hacking.
>
>
>
> #####
> Ongoing rebuttal flow:
> #3 - Generalization
>
> We thank the reviewers for raising this concern. To assess whether our results generalize beyond the validation set, we evaluated the RL-trained model on the held-out test set (409 samples), obtaining the same score of 64% on both splits. While this consistency is encouraging, we acknowledge that it may partly reflect limited diversity in the dataset splits themselves, since FHIR-AgentBench questions are programmatically generated from a fixed set of clinically relevant templates rather than drawn from a broad distribution of real-world queries.
> Regarding schema generalization more broadly: we do observe multi-turn self-correction behavior in our agent (e.g., recovering from malformed Python or unexpected API responses), which provides a basis for adapting to unfamiliar schemas at inference time. However, systematically evaluating robustness to schema variation would require constructing new datasets that model common sources of divergence across FHIR deployments: a significant independent effort that falls outside the scope of this paper.
> We fully agree that schema robustness is a prerequisite for real-world clinical adoption, and we explicitly position this as an important direction for future work. We will make this limitation and its implications clearer in the revised manuscript.

---

> > ### Author Rebuttal · Reviewer_8uzj · 2026-04-04
> >
> > The additional experiments meaningfully strengthen the paper. Some concerns remain partially open. The confounded design between Experiments 1 and 2 is not ablated, and comparisons against supervised fine-tuning and other open-weight base models would further clarify the contribution. But I will raise my score accordingly.

---

> > > ### Author Response · Authors · 2026-04-08
> > >
> > > Thank you 8uzj for acknowledging the additional experiments and the discussion. In addition to SFT-ablations (also requested by other reviewers), we are adding a same-scale open-source coder baseline. Thank you!

---

### Official Review · Reviewer_fkDD · 2026-03-13

**Soundness:** 2
**Presentation:** 3
**Significance:** 2
**Originality:** 2
**Overall Recommendation:** 4
**Confidence:** 4

**Summary:**

This paper addresses the problem of question answering over FHIR-formatted electronic health records, framing it as a sequential decision-making problem over a typed, directed resource graph. The authors implement a multi-turn CodeAct agent that issues executable Python programs to query a FHIR server and process returned JSON, then post-train this agent using Group Relative Policy Optimization (GRPO) with binary correctness rewards from an LLM judge. The base model is Qwen2.5-Coder-Instruct-7B, and training is conducted within the SkyRL framework on the FHIR-AgentBench benchmark. Two experiments are presented, differing in normalization scheme (standard sequence-mean vs. Dr. GRPO constant normalization), system prompt design, and learning rate. The best configuration achieves 64% answer correctness on the 424-question validation set, up from 50% reported for o4-mini with multi-turn code in the original benchmark paper. The authors provide a per-resource-type error analysis and show that targeted prompt changes substantially improve performance on Medication/MedicationRequest questions.

**Compliance With Llm Reviewing Policy:**

Affirmed.

**Final Justification:**

The rebuttal addressed my two main concerns: the base model ablation (23.6% vs. 64% after RL) cleanly isolates RL's contribution, and the judge validation (96.2% human agreement) confirms reward reliability. The cleaner Qwen3-8B setup and per-resource failure analysis further strengthen the paper. Originality remains limited since all components are from prior work, and an SFT baseline is still missing. However, a 7B model outperforming frontier closed models on realistic clinical QA has clear practical value for privacy-constrained healthcare settings. I raised my score from 3 to 4 and encourage the authors to add SFT and open-weight comparisons in the camera-ready.

**Key Questions For Authors:**

1. What is the accuracy of the base Qwen-7B model (before any RL training) when evaluated with the Experiment 2 system prompt? This would isolate the contribution of RL post-training from prompt engineering. If the base model with the revised prompt already reaches 55–60%, the value added by RL training is much smaller than suggested.

&nbsp;

2. Have you validated the LLM judge against human correctness annotations on a representative sample of the 424 questions? If the judge has, say, 85% agreement with humans, the reported 64% accuracy has substantial measurement uncertainty. Knowing the judge's precision and recall for both correct and incorrect answers would help calibrate the results.

&nbsp;

3. Can you ablate the three changes between Experiment 1 and Experiment 2 (normalization, prompt, learning rate) individually? The current presentation makes it impossible to identify which factor drives the 4-percentage-point improvement, and a controlled ablation would strengthen the experimental contribution considerably.

&nbsp;

4. How does the trained agent perform when the FHIR server uses a different profiling convention or when optional fields are populated differently? Even a synthetic perturbation experiment (e.g., removing certain fields or renaming extensions) would provide evidence about the robustness claim that motivates the paper.

**Limitations:**

The authors acknowledge several limitations, including the simplicity of the retrieval interface (Section 5), the insufficiency of 64% accuracy for clinical deployment, and the need for clinician oversight. However, they do not discuss the risk of the agent producing plausible but incorrect clinical answers that could mislead downstream users, nor do they address potential biases in the MIMIC-IV patient population. The societal impact statement (page 9) is generic and would benefit from concrete discussion of failure consequences in clinical settings.

**Strengths And Weaknesses:**

**Strengths**

- A central concept presented by this manuscript is that FHIR graph traversal for clinical QA is a first-class sequential decision-making problem, and the paper makes a convincing case for why this matters. The introduction (Section 1) and background (Section 2.1) clearly articulate how schema heterogeneity, optional fields, and site-specific profiling make single-shot prompting brittle. Framing retrieval over a typed directed graph as a partially observable decision process (Section 3.1) is well-motivated and connects the healthcare interoperability challenge to the RL and agent-learning communities in a natural way.

- The headline result of improving from 50% (o4-mini) to 64% using a 7B open-weight model is meaningful for the healthcare domain, where privacy constraints often preclude closed-API deployment. This is demonstrated clearly in Figure 4 and discussed in Section 4.2. The fact that a substantially smaller model can outperform a frontier closed model after RL post-training provides a credible proof of concept for on-premise clinical agents.

- The per-resource-type breakdown in Figure 4 and the qualitative analysis of MedicationRequest-to-Medication reference-following failures (Section 4.2) go beyond aggregate accuracy and provide actionable diagnostic insights. The observation that the o4-mini baseline and the Experiment 1 agent both fail to follow MedicationRequest references to the Medication resource, and that prompting the agent to inspect resource structure before coding substantially mitigates this, is a concrete and informative finding. The trajectory examples in Appendix A.5 and Figure 5 further support this analysis.

- A significant problem examined by the study is the engineering challenge of building an RL post-training pipeline for multi-turn tool-using agents over structured data. The paper describes the full system, including the BLAZE FHIR server setup, the SkyRL integration, the LLM judge reward, and the XML-tagged action format (Section 3.2). The system prompt in Appendix A.3 and the trajectory viewer in Figure 9 are useful practical artifacts.

---

**Weaknesses**

- The two experiments differ simultaneously in normalization scheme (sequence-mean vs. Dr. GRPO), system prompt (three in-context examples vs. one example with explicit inspection instructions), and learning rate (1e-6 vs. 3e-6). Because all three factors change at once, it is impossible to attribute the improvement from Experiment 1 (60%) to Experiment 2 (64%) to any single factor. The paper acknowledges the prompt change and normalization change but does not ablate them. Without controlled comparisons, the relative contribution of RL training itself versus prompt engineering remains unclear. For instance, one cannot tell from the current results whether the revised system prompt alone, applied to the base model without any RL training, would already close much of the gap.

- All results are reported on a single benchmark split of 424 questions from FHIR-AgentBench. There is no evaluation on a held-out test set, no cross-validation, and no confidence intervals or variance across multiple training seeds. Several natural baselines are absent, including the base Qwen-7B model before RL training (only the learning curves starting point can be inferred, around 25–40%), other open-weight models of comparable or larger size, and supervised fine-tuning on trajectories. The comparison to o4-mini uses numbers from the original benchmark paper rather than a controlled re-evaluation under the same judge, which introduces potential inconsistency in evaluation conditions.

- The individual components, including CodeAct-style agents, GRPO training, LLM-as-judge reward, and Dr. GRPO normalization, are all drawn from recent prior work (Wang et al. 2024; Shao et al. 2024; Yu et al. 2025; Liu et al. 2025a). The paper applies these components to FHIR QA but does not introduce new algorithmic ideas, new reward shaping specific to graph traversal, or new theoretical insights about when RL post-training helps for structured data agents. The closest methodological novelty is the system prompt design encouraging schema inspection, which is more of an engineering insight than a research contribution.

- The reward signal comes from Qwen2.5-Instruct-32B acting as an LLM judge (Section 3.2). The paper does not validate the judge's accuracy against human annotations, nor does it report inter-annotator agreement or judge error rates. Since the judge provides the training reward and the evaluation metric, any systematic bias in the judge (for example, leniency toward certain answer formats or difficulty with numerical precision) could inflate reported accuracy and distort training. The evaluation protocol follows Lee et al. (2025), but the original paper's judge may have been a different model.

- The experiments use 100 MIMIC-FHIR demo patients, which represents a small and homogeneous subset of the MIMIC-IV database. The paper motivates the work by emphasizing cross-site schema variability (Section 2.1), but all training and evaluation occur on a single FHIR deployment with a single profiling convention. Whether the learned policy transfers to different FHIR servers, different profiling choices, or different patient populations is not addressed. This substantially limits the strength of the claim that RL post-training improves robustness to schema variability.

---

> ### Author Rebuttal · Authors · 2026-03-31
>
> Thank you reviewer fkDD for the feedback and the questions, especially for pointing out the “engineering challenge of building an RL post-training pipeline for multi-turn tool-using agents over structured data” with multiple components (blaze, Judge, etc). The criticism is valid, and we address it in more detail.
>
> 1. Accuracy is 23.6% for the Qwen3-8B base model. After RL-training, it is 64%. Please see response to reviewer yPmE for more information.
> 2.Please see response below (#2 Judge). In short: human and Qwen Judge agree to 96% on the 424 validation set questions. Precision & recall both above 97%.
> 3. Ablations: we revised our experimental design (now native tool calling, native reasoning, Qwen3) and re-run some controlled experiments (8B GRPO baseline, +clip higher, 4B GRPO baseline). Clip higher does not seem to contribute to accuracy. Surprisingly, the 4B model after RL training scores above 50%, matching closed source models (e.g. our Gemini3-Flash or o4-mini baseline).
> 4. We talk about this also in the response to reviewer yPmE. In short: While observing “self correction” in multi turn tool call (e.g. when Python code is wrong) which should generally be the basis for adapting to different schemas on the fly, unfortunately it was out of scope of this paper to control for this. We leave this for a future study more on the data-side, while we understand this paper as more of an engineering / framework contribution which tests multi-turn agents for FHIR retrieval.
>
> ____________________________________________
> #2 - Validation of Judge
>
> We agree that the judge required additional validation. In response, we performed a direct agreement study between the automated judge and human (n=1 board certified physician) annotations on the 424 validation samples.
>
> The resulting agreement numbers are:
> - Human vs. Qwen-70B: 96.2% agreement (precision=96.7%, recall=97.4%), matching the agreement between human and Judge in Lee et al. 2025 (97%).
> - Gemini vs. Qwen-70B: 93% agreement, where most (24 of 30) disagreements are attributable to Gemini being less forgiving in questions without an answer. (non-empty questions have 97%-100% agreement). The Qwen Judge is more correct according to human judgement.
>
> We used two judges for robustness. Qwen2.5-72B was used in the internal training / evaluation workflow, while the Gemini judge serves as an external cross-check and is used in training on “normal” compute. This is practically important in our setting because parts of the infrastructure are tied to clinical data handling constraints, and we wanted an additional, independent signal. We will clarify this motivation in the revision.
>
> We also note that the two judges differ in strictness. In particular,Gemini judge was noticeably harsher on partially correct or underspecified answers. We will report this explicitly and frame the results accordingly, rather than presenting the judge as a perfect proxy for human evaluation.

---

> > ### Author Rebuttal · Reviewer_fkDD · 2026-04-02
> >
> > I thank the authors for the substantial revisions and new experiments. The base model comparison (23.6% vs. 64% after RL) convincingly isolates the contribution of RL post-training, and the judge validation (96.2% human agreement) addresses my earlier concern. The transition to a cleaner experimental setup with Qwen3-8B and native tool calling also improves interpretability. I am raising my score from 3 to 4, as the revised results make a credible case that RL post-training meaningfully improves multi-turn reasoning over FHIR graphs. I still encourage the authors to include an SFT baseline and additional open-weight model comparisons in the camera-ready version.

---

> > > ### Author Response · Authors · 2026-04-08
> > >
> > > Thank you fkDD for the feedback!
> > > We are adding an ablation to address your request for a SFT baseline: SFT-only / SFT+RL. The paper will be updated accordingly.
> > > We will also provide final model weights and a harness to reproduce the results.

---

### Official Review · Reviewer_yPmE · 2026-03-13

**Soundness:** 2
**Presentation:** 1
**Significance:** 2
**Originality:** 1
**Overall Recommendation:** 3
**Confidence:** 4

**Summary:**

The paper builds a multi-turn CodeAct agent with access to a FHIR retrieval tool and a persistent Python interpreter, then post-trains the agent with GRPO-style RL using an LLM judge as the reward source.

**Compliance With Llm Reviewing Policy:**

Affirmed.

**Key Questions For Authors:**

Please address points 2-4 and 7 in the Strengths And Weaknesses section.

**Limitations:**

yes

**Strengths And Weaknesses:**

1. The paper addresses a real and important problem. FHIR is increasingly central to interoperable healthcare data, and reasoning over FHIR is genuinely difficult because useful answers often require multi-hop traversal across heterogeneous resources rather than simple table lookup. Framing this as sequential tool-using reasoning is sensible and well motivated.

2. The main comparison is not clean, so the source of improvement is hard to isolate. The headline result compares the proposed RL-trained Qwen-7B system against a previously reported o4-mini baseline from FHIR-AgentBench. But the proposed system differs from that baseline in multiple ways at once: different base model, different prompting, explicit early-inspection encouragement, XML tool schema, and RL post-training. Because many ingredients change simultaneously, it is hard to attribute the gain specifically to RL. A more convincing evaluation would include strong ablations against the same Qwen base model with no RL, with prompt-only changes, and with comparable search/exploration incentives.

3. The reward signal is weak and potentially brittle. The trajectory reward is binary: the answer must match the expected format and be judged correct by a Qwen2.5-32B LLM judge, otherwise reward is zero. That is simple, but for long-horizon tool-use behavior it is a coarse signal. The paper does not provide evidence that the judge is reliable on partial-credit cases, numerically close answers, or answers supported by correct retrieval but phrased differently. Since the judge is central to training, stronger validation of reward quality is needed.

4. The evaluation remains quite narrow. All results are reported on the 424-question validation set over 100 MIMIC-FHIR demo patients. That is useful as a first benchmarked result, but it is a limited experimental setting for the paper’s broader claims about robustness to heterogeneous FHIR deployments.

5. The technical novelty is limited. The paper combines known ingredients: CodeAct-style agent execution, GRPO-style RLVR, and a domain-specific FHIR environment. It is more a careful domain adaptation of existing RL-for-tool-use ideas than a fundamentally new learning method.

6. Writing and presentation need polishing. There are multiple minor language and presentation issues such as awkward phrasing, inconsistent capitalization, some typographical errors (“practioners”), and equations/explanations that could be clearer.

7. The paper’s central claim about enforcing data-integrity constraints is under-evidenced. The abstract says the method improves performance while enforcing data-integrity constraints, and the text emphasizes traversal correctness. But I did not see a dedicated quantitative analysis of constraint-violation rates before and after RL, beyond answer correctness and resource-type failure interpretation. If constraint adherence is a core contribution, it should be directly measured.

---

> ### Author Rebuttal · Authors · 2026-03-31
>
> Dear reviewer yPmE,
>
> Thank you for the valuable feedback. After contextualizing other reviewers' replies, we identified four major concerns shared by all:
>
> Confounded experiment design (mentioned by yPmE, fkDD, 8uzj, CC6T): In our two experiments, it is unclear which changes contributed to the increase in answer correctness (normalization, prompting, …)
> Validation of Judge (yPmE, fkDD, 8uzj): Judge was not validated against human correctness etc.
> Generalization (yPmE, fkDD, CC6T): Only 424 questions in validation set / ability to generalize to new schemas.
> More / better baselines to understand the contribution of RL (yPmE, fkDD, 8uzj, CC6T): Qwen baselines missing without RL, but using our harness
>
> We like to thank all reviewers for their questions, comments, and concerns.
>
> We are glad the reviewers [yPmE, fkDD, 8uzj, CC6T] agree with identifying FHIR resources as an important domain for agents. Pioneering work has been done by Lee et al. 2025 with the creation of FHIR-AgentBench. This allows for a principled comparison between the abilities of different LLM in medical QA over knowledge graphs. To move from benchmark towards production there are severe issues to overcome, namely performance and privacy constraints. To address the latter; open weight models are an increasingly popular method. To boost performance we decide to finetune it further. This changes the role of the benchmark from evaluation to training system. Our core contribution is modification of the Lee et al. 2025 framework into a principled FHIR-Agent training infrastructure.
> Doing so we show infrastructure choices (Blaze), design of the reward signal (judges), connection of the model with the tools through a harness as well as performance gains from performing finetuning in our training infrastructure on the example of the open weight model Qwen3-8B (note that this is the instruction tuned version of Qwen3-8B, the basemodel is Qwen3-8b-Base).
>
> Addressing the major concerns raised, we have substantially revised our experimental setup and re-run all experiments accordingly. Our goal was methodological clarity. We therefore transitioned to Qwen3-8B (replacing Qwen2.5-7B), adopting both native tool calling and native reasoning. Concretely, native tool calling means we rely exclusively on Qwen3's built-in tool calling format via the Hermes parser. Similarly, rather than engineering prompts that force the model to externalize its reasoning into XML tags, we now leverage Qwen3's native reasoning mechanism directly. Since our primary contribution is not prompt engineering, we have also substantially reduced prompt complexity, removing few-shot examples entirely. The result is a significantly cleaner and more principled experimental setup, which better isolates the model's intrinsic capabilities.
>
>
> #1 - Confounded experiment design
> We agree that, in the original submission, multiple changes were introduced together, making it hard to attribute improvements to any one factor. To address this, we redesigned the setup so that the agent interface is substantially cleaner and closer to the model’s native operating mode.
> Specifically, in the revised experiments we use:
> Qwen3 8B instead of Qwen2.5 7B,
> native tool calling via the model’s standard tool-calling format (also the closed model baselines now use their native tool calling format)
> native reasoning, rather than prompting the model to emit reasoning inside XML tags,
> a substantially shorter system prompt without few-shot examples:
>
> You are a FHIR data analyst. Answer patient data questions by querying a FHIR server. Rules:
> Every claim must trace to a print() output or computation.
> If unsure about a resource's schema, print a sample first.
> Keep your reasoning brief.
> When done, call finish.
>
> This reduces the number of moving parts and makes the comparison more interpretable. We also added the following controlled comparisons (always accuracy):
> o4-mini with the new simplified system prompt: 46.1%
> o4-mini with the original FAB system prompt (Lee et al. 2025): 50%
> Gemini3-flash with our prompt: 49.76%
> GLM-5 with our prompt:
> Qwen3 8B base model in our harness, no RL (temp=0.1): 21.7%
> Qwen3 8B base model in our harness, no RL (capped at 12 turns, temp=0.1): 23.6%
> Qwen3 8B after RL training (temp=0.1): 64%
> These revised comparisons are intended to separate the contributions of prompting, harness design, and RL more clearly. In the camera-ready version, we will make this distinction explicit in the experimental section and discussion.

---

> > ### Author Rebuttal · Reviewer_yPmE · 2026-04-04
> >
> > The rebuttal does not satisfactorily address my main concerns. In particular, it replaces the original setup with a new Qwen3-based configuration rather than isolating the confounds in the submitted system, does not provide a proper validation of the LLM judge, does not meaningfully expand the evidence for generalization beyond the narrow benchmark setting, and still lacks direct quantitative evaluation of the claimed data-integrity constraints. Therefore, I still think that the paper does not cross the bar for acceptance and keep my original score.

---

> > > ### Author Response · Authors · 2026-04-07
> > >
> > > Dear reviewer yPmE,
> > > Thank you for the follow-up. Our rebuttal answers were distributed in the “ongoing rebuttal flow” section (denoted by the trailing sections: #1, #2, #3, #4) to each reviewer independently, although they are relevant to all. We realize this is not ideal for reading flow, and we apologize for missing to address your point 7 entirely.
> > >
> > > Point 2: confounded experiment design / unclear source of improvement.
> > > We agree that the originally submitted comparison changed too many factors at once: prompting, tool interface, training parameters. Therefore, we re-ran the study in a substantially more controlled setup. This new setup is not intended to retroactively deconfound the original system, but to answer the attribution-question more directly.
> > > In the revised setup, we keep the FHIR environment and harness fixed and compare the same Qwen3-8B agent before and after RL, while simplifying the interface to native tool calling and a short system prompt. Under this controlled comparison:
> > > Qwen3-8B in our harness, no RL: 21.7%
> > > Qwen3-8B in our harness, no RL, 12-turn budget: 23.6%
> > > Qwen3-8B after RL: 64.0%
> > > These results support a narrower but cleaner conclusion compared to the original submission: within the same agent setup, RL contributes a large gain beyond merely placing the base model in the harness. We agree that the originally submitted Qwen2.5-based result did not isolate this sufficiently, and we revised the paper to make that distinction explicit.
> > >
> > > Point 3: reward signal / judge reliability.
> > > We agree that the judge required direct validation. In response, we evaluated our primary judge, Qwen2.5-72B, against physician annotation on the full 424-sample validation set. The agreement is 96.2%, with precision 96.7% and recall 97.4%. Outside critical compute infrastructure we use a Gemini-3.1-Flash-Lite-Preview judge. Agreement between Qwen2.5-72B and Gemini-3.1-Flash-Lite-Preview is 93.0%. The disagreements are concentrated in no-answer cases, where Gemini-3.1-Flash-Lite-Preview is typically stricter.
> > >
> > > Point 4: narrow evaluation / limited generalization evidence.
> > > We agree with this concern. The current paper does not establish robustness to heterogeneous FHIR deployments or unseen schema variants. The strongest additional evidence we can provide is that the RL-trained model achieves the same 64.0% accuracy on the held-out 409-sample test set, matching the validation result while using the same FHIR schema. This supports generalization across benchmark splits, but it does not by itself demonstrate schema-shift robustness. We will therefore narrow the manuscript’s wording accordingly. Our claim is benchmark-level generalization within FHIR-AgentBench, not robustness to arbitrary deployment-specific FHIR variations. We agree that evaluating schema variation across institutions is important future work and should be framed as such, not implied as already solved.
> > >
> > > Point 7: data-integrity constraints.
> > > You are right that our wording here was imprecise. What we intended to communicate: in clinical settings, privacy, governance, and data-locality constraints often limit the use of external closed APIs on patient data, which makes open-weight models and local post-training practically important. This is a motivation for the work, not a separate quantitative result, and we revised the abstract and body text to make that distinction explicit.
> > >
> > > The paper shows that in a clinically relevant FHIR QA environment, a CodeAct-style open-weight agent can be improved substantially through RL post-training over the same base model and harness, using a reward signal from Qwen2.5-72B that agrees strongly with physician annotation. We believe this contribution is practically significant: FHIR is the dominant interoperability standard in modern healthcare IT, and to our knowledge this is among the first demonstrations of RL post-training applied to FHIR. The ability to fine-tune open-weight models locally - without routing patient data to external APIs - directly addresses a deployment constraint that affects nearly every real-world clinical setting. We therefore see this work as establishing an important and reproducible baseline for a class of problems that will only grow in relevance as FHIR adoption continues to expand globally.
> > >
> > > Thank you again for pushing us to sharpen both the experimental claim and the wording of the paper.

---

### Decision · Program_Chairs · 2026-04-30

**Decision:**

Accept (regular)

**Comment:**

This paper addresses question answering over FHIR (Fast Healthcare Interoperability Resources), the dominant standard for interoperable clinical data exchange. The authors frame FHIR retrieval as a sequential decision-making problem over a typed, directed resource graph. They implement a multi-turn CodeAct agent that generates executable Python programs to query a FHIR server and process returned JSON. The agent is post-trained using GRPO-style reinforcement learning with binary correctness rewards from an LLM judge (Qwen2.5-72B).

The remaining issues after rebuttal includes: 1) Missing supervised fine-tuning (SFT) baseline to test whether gains come from imitation learning vs. RL specifically. 2) Original submission changed three factors simultaneously (normalization, prompt, learning rate), making attribution impossible. Redesign during rebuttals helps but original confounding in submission remains. 3) Narrow evaluation scope with limited generalization evidence.

This paper received a mixed evaluation with three WA and one WR. The main weakness is the weak technical novelty. All core components from prior work: CodeAct (Wang et al. 2024), GRPO (Shao et al. 2024), Dr. GRPO normalization (Liu et al. 2025a), LLM-as-judge (Lee et al. 2025). Main novelty is domain adaptation to FHIR and system engineering (harness design, reward integration). Thus I would recommend a weak acceptance but would not mind if it is rejected.